# Manufacture of a Stable Lyophilized Formulation of the Live Attenuated Pertussis Vaccine BPZE1

**DOI:** 10.3390/vaccines8030523

**Published:** 2020-09-13

**Authors:** Marcel Thalen, Anne-Sophie Debrie, Loic Coutte, Dominique Raze, Ken Solovay, Keith Rubin, Nathalie Mielcarek, Camille Locht

**Affiliations:** 1ILiAD Biotechnologies, New York, NY 10003, USA; marcel@iliadbio.com (M.T.); ken@iliadbio.com (K.S.); keith@iliadbio.com (K.R.); 2Centre d’Infection et d’Immunité de Lille, Univ. Lille, CNRS, Inserm, CHU Lille, Institute Pasteur de Lille, U1019–UMR9017–CIIL–Center for Infection and Immunity of Lille, F-59000 Lille, France; anne-sophie.debrie@ibl.cnrs.fr (A.-S.D.); loic.coutte@inserm.fr (L.C.); dominique.raze@ibl.cnrs.fr (D.R.); nathalie.mielcarek@inserm.fr (N.M.)

**Keywords:** pertussis, live attenuated vaccine, lyophilization, vaccine stability

## Abstract

Current pertussis vaccines protect against disease, but not against colonization by and transmission of *Bordetella pertussis*, whereas natural infection protects against both. The live attenuated vaccine BPZE1 was developed to mimic immunogenicity of natural infection without causing disease, and in preclinical models protected against pertussis disease and *B. pertussis* colonization after a single nasal administration. Phase 1 clinical studies showed that BPZE1 is safe and immunogenic in humans when administered as a liquid formulation, stored at ≤−70 °C. Although BPZE1 is stable for two years at ≤−70 °C, a lyophilized formulation stored at ≥5 °C is required for commercialization. The development of a BPZE1 drug product, filled and lyophilized directly in vials, showed that post-lyophilization survival of BPZE1 depended on the time of harvest, the lyophilization buffer, the time between harvest and lyophilization, as well as the lyophilization cycle. The animal component-free process, well defined in terms of harvest, processing and lyophilization, resulted in approximately 20% survival post-lyophilization. The resulting lyophilized drug product was stable for at least two years at −20 °C ± 10 °C, 5 °C ± 3 °C and 22.5 °C ± 2.5 °C and maintained its vaccine potency, as evaluated in a murine protection assay. This manufacturing process thus enables further clinical and commercial development of BPZE1.

## 1. Introduction

Pertussis or whooping cough is a highly contagious respiratory disease that can be fatal, especially in young infants [1]. However, it also affects adolescents and adults [2]. Its main causative agent, *Bordetella pertussis*, is a Gram-negative coccobacillus that effectively colonizes the respiratory tract, but usually does not disseminate to other organs [3]. Currently available vaccines are effective in controlling the disease, but do not prevent infection by, nor transmission of *B. pertussis* [4,5]. Furthermore, vaccine-induced immunity wanes rapidly, especially after administration of the second-generation acellular vaccines [6] now used in most industrialized countries.

In contrast to vaccination, prior infection by *B. pertussis* prevents secondary infections and induces long-lasting, although not life-long immunity [7]. Based on these observations, a vaccination strategy to mimic the immunogenicity of natural infection without causing disease was developed by using the live attenuated nasal pertussis vaccine BPZE1. This vaccine candidate is an attenuated *B. pertussis* Tohama I derivative in which three major toxins, pertussis toxin (PT), dermonecrotic toxin (DNT) and tracheal cytotoxin, have been genetically inactivated or removed [8].

In non-human primates, a single administration of BPZE1 protected against both pertussis disease and colonization by a highly pathogenic *B. pertussis* clinical isolate [9]. Recent studies in mice have shown that BPZE1 induces potent mucosal IgA and IL-17-producing tissue-resident CD4^+^ memory T cells in the nose, which appears to be crucial for long-term protection against nasal colonization by *B. pertussis* [10]. BPZE1 is now in clinical development and has already successfully completed two phase I studies, which have shown that the vaccine is safe in adult volunteers, and is able to transiently colonize the human nasal cavity and to induce antibody responses to *B. pertussis* antigens [11,12].

However, during previous non-clinical and clinical studies, a liquid formulation of BPZE1 was used. The bacteria were cultured in shake flasks and had to be stored at −70 °C to maintain BPZE1 viability, a major impediment to commercialization. In this study, a scalable production process was developed based on growing BPZE1 in a bioreactor, formulating the culture in a suitable lyophilization buffer, followed by lyophilization in single-dose vials. Real-time stability studies indicated a shelf life of at least two years.

## 2. Materials and Methods

### 2.1. Bacterial Strains and Growth Conditions

Virulent *B. pertussis* BPSM [13] and BPBCTA1 [10,14] were grown at 37 °C on Bordet–Gengou (BG) agar containing 100 µg/mL streptomycin and 10 µg/mL gentamycin (for BPCTA1) and supplemented with 1% glycerol and 10% defibrinated sheep blood, as described [8]. After growth, the bacteria were harvested by scraping the plates and resuspended in phosphate-buffered saline (PBS) at the desired density. The BPZE1 vaccine strain [8] Working Cell Bank (WCB) was grown in fully synthetic Thijs medium [15] under agitation. After addition of 20% vol/vol of 86% glycerol, aliquots were filled in 1.5-mL cryo-vials and the WCB was stored at −70 °C until further use, as described [11,12].

### 2.2. Preparation of the BPZE1 Drug Substance

The WCB, with a volume of 1.5 mL, was inoculated in 125-mL Erlenmeyer flasks containing 28.5 mL of Thijs medium [16], a chemically defined medium optimized for virulence gene expression, derived from the Stainer Scholte medium [17]. The second pre-culture, consisting of a 2-L Erlenmeyer flask with 0.5 L Thijs medium, was inoculated at an OD_600_ of 0.1, which was in turn used to inoculate, at an OD_600_ of 0.1, 5 × 2-L flasks containing 0.5 L Thijs medium each. The 5 cultures were pooled and added to the 50-L bioreactor (Sartorius, 50 L SUB) with 20 L Thijs medium so that the bioreactor started at an OD_600_ of 0.1. The fermentation was performed at 35 °C, dissolved oxygen was fixed at 20% using compressed air supplied through the sparger, and the pH was maintained at pH 7.5 using 0.2 M lactic acid. All product contact materials, such as the culture and medium flasks, containers, tubing, filters, connectors, as well as the bioreactor, were single use. After reaching the target OD_600_ of 1.1–1.4, a sample of 8 L culture was concentrated and/or diafiltered to an OD_600_ as specified, using hollow fiber tangential flow filtration (TFF; 750 kDA mPES membrane 1400 cm^2^, Spectrum) at a maximum transmembrane pressure of 0.3 bar.

### 2.3. Lyophilization of BPZE1

Initial culture and lyophilization development resulted in a lyophilization buffer and lyophilization cycle at a small scale. For all larger scale cultures the lyophilization buffer, cooled at 4 °C, was added in a 1:1 ratio to the bacterial suspension, using 100 g/L sucrose as the main cryoprotectant. The resulting formulated drug substance was then filled in 1-mL aliquots in DIN 2R vials with a 13-mm bromobutyl lyophilization stopper and was lyophilized using a conservative cycle, consisting of primary drying at −34 °C at 100 microbar until the Pirani and the capacitance manometer readings converged, indicating that sublimation had ended. Primary drying was followed by a ramp from −34 to 30 °C in 12 h, followed by holding the temperature at 30 °C until the pressure increased less than 10 microbars after closing the valve to the condenser chamber, indicating that the product was dry. After stoppering, the vials were cooled to 4 °C until unloading, followed by capping the vials with an aluminum cap.

### 2.4. Plate Count

The enumeration of the colony forming units (CFU) was performed by plating 1, 2 and 5 times 10-fold dilutions of drug substance or drug product samples on BG agar plates supplemented with 15% sheep blood. All dilutions were plated in triplicate so that on average 9 plates were counted to obtain a single result. The specification of the drug product after lyophilization was set to 0.2–4.0 × 10^9^ CFU/mL.

### 2.5. Microbial Safety Tests on Drug Substance and Drug Products

The absence of *Staphylococcus aureus*, *Pseudomonas aeruginosa* and bile-tolerant organisms was tested according to the United States Pharmacopoeia, test 1111 (USP<1111>), whereas the purity of both the drug substance and the drug product was tested according to USP<61> and USP<62> [18]. All runs complied with all USP safety tests.

### 2.6. Mouse Colonization and Potency Assays

All the animal experiments were carried out in accordance with the guidelines of the French Ministry of Research regarding animal experiments, and the protocols were approved by the Ethical Committees of the Region Nord Pas de Calais and the Ministry of Research (agreement number APAFIS#9107 ± 201603311654342 V3). BALB/c mice were purchased from Charles River Laboratories (l’Abresle, France) and kept at the Institut Pasteur de Lille animal facility under specific pathogen-free conditions. For colonization assays, the lyophilized drug product was resuspended in distilled water, and the various BPZE1 suspensions were diluted in PBS to 10^5^ CFU per 20 µL and were directly nasally administered to six-week old BALB/c mice. The mice were sacrificed 3 h, 24 h or 3 days after infection, and nasal homogenates were prepared as described [10] and then plated in ten-fold serial dilutions onto BG blood agar plates and incubated at 37 °C for 3–5 days to quantify colonization by CFU counting. To determine the potency of the various BPZE1 formulations, six-week old BALB/c mice were intranasally vaccinated with 10^5^ CFU of BPZE1 or received PBS intranasally, as described [19]. Four weeks later, the mice were challenged intranasally with 10^6^ CFU of virulent BPBCTA1. Lung colonization was determined 3 h and 7 days post-challenge as described above.

### 2.7. Genetic Stability Assays

The genetic stability of the various BPZE1 preparations was evaluated by polymerase chain reaction (PCR) targeting the *dnt* and *ampG* genes, as described [20]. The PT S1 subunit gene *ptxA* was analyzed by quantitative PCR (qPCR) for the absence of reversion of the two codon changes introduced to inactivate PT [8]. Approximately 10^10^ CFU of the BPZE1 preparations were harvested by centrifugation and suspended in buffer B1 (50 mM Na2EDTA, 50 mM Tris-HCl, 0.5% Tween 20, 0.5% Triton X100, pH 8.0; Qiagen, #19060, Hilden, Germany), containing RNaseA and proteinase K, and incubated at 37 °C for 30 min. The bacteria were then lysed in lysis buffer for 30 min at 50 °C and applied to a Qiagen genomic-tip 100/G column. After washing and elution as recommended by the manufacturer, the DNA was precipitated with isopropanol (ACROS, Geel, Belgium), centrifuged at 5000× *g* for 15 min, washed with ice-cold 70% ethanol, air dried for 10 min and resuspended in 100 µL bi-distilled water. The DNA concentration was measured using a NanoDrop 2000 c spectrophotometer. One µL of BPZE1, BPSM or BPSM-spiked BPZE1 DNA corresponding to 10^7^ genome copies was mixed with 19 µL of LightCycler 480 SYBR Green I Master mix containing 0.5 µM of primer pairs in 96-well LightCycler 480 plates. The plates were sealed with specific plastic film, transferred to the LightCycler 480 and subjected to 15 min incubation at 95 °C, followed by 1 to 40 cycles of denaturation for 15 s at 95 °C, annealing for 8 s at 68 °C and 18 s of elongation at 72 °C. The data were then analyzed using the LightCycler 480 software release 1.5.0. To ensure that the assay would be able to detect one potential reversion among 10^6^ copies of genome, 10 copies of BPSM genomic DNA were mixed with 10^7^ copies of BPZE1 genomic DNA. All primers were purchased from Eurogentec (Liège, Belgium), and their sequences are provided in Table 1.

## 3. Results

### 3.1. BPZE1 Drug Substance Development

The conditions for the fermentation in shake flasks and the production bioreactor were essentially as described [21], with minor modifications as described in the Materials and Methods section. The development work mostly focused on determining the culture conditions and lyophilization buffer composition that resulted in a homogenous drug substance suspension and maximal survival after lyophilization. *B. pertussis* produces a number of virulence factors that enable binding to epithelial cells as well as to each other, and is capable of biofilm formation. In a bioreactor, biofilm formation leads to bacterial clumping and therefore to an inherently inhomogeneous vaccine drug product. Clumping in the bioreactor can be avoided by increasing agitation, but too high shear forces during fermentation or ultrafiltration lead to cell damage, which translates into low survival after lyophilization. In the 20-L bioreactors with 8 L medium, run at 400 RPM using a six-blade Rushton impeller, post-lyophilization survival did not exceed 45%, whereas in the 50-L bioreactor with 20 L medium, run at 150 RPM using a three-blade marine impeller showed post-lyophilization survival of up to 65% under similar conditions (data not shown).

At the 8-L bioreactor scale, the suspension at OD_600_ of 0.5 showed little clumping, but poor survival after lyophilization compared to OD_600_s of >1.0. Therefore, all the subsequent cultures were harvested at an OD_600_ of 1.1–1.6. This OD_600_ corresponds to approximately 50% to 80% of the maximum OD_600_, well before all medium substrates were consumed, so that the bacteria were in a state that would result in high survival after lyophilization. In order to halt cellular metabolism during the period between the harvest and freezing on the shelf of the lyophilizer, the addition of cold lyophilization buffer was found to be suitable.

To minimize the impact of bioreactor and TFF geometry on post-lyophilization survival, all 50-L bioreactors were run using the same conservative conditions during fermentation and ultrafiltration, compromising between minimizing shear stress and avoiding clumping.

### 3.2. Lyophilization Buffer Development

The manufacturing process development for the drug product consisted of developing a lyophilized formulation, including a lyophilization buffer and a matching lyophilization cycle, as well as verifying that the developed process did not interfere with the biological activity of the BPZE1 drug product. It is especially important that the drug product maintains its ability to reduce the bacterial burden in the lungs by at least two orders of magnitude in the murine protection assay. The target drug product attributes are shown in Table 2.

The formulation of the lyophilization buffer was based on commonly used cryoprotectants, containing 5% to 10% sucrose or trehalose, sometimes in combination with other cryoprotectants such as hydroxy ethyl starch (HES) or sodium glutamate (MSG). A single bacterial suspension was used to generate all formulations shown in Table 3.

All formulations showed a residual moisture content (RMC) below the 2.5% target and a glass transition temperature (Tg) above the 35 °C target. Sucrose appeared superior over trehalose as a cryoprotectant when used alone. The addition of HES, MSG or both to 5% or 10% of trehalose or sucrose appeared to reduce, rather than enhance post-lyophilization survival, except for the addition of 7% HES to 10% sucrose, which yielded 44% survival compared to the 38% survival using sucrose by itself. However, since the variability of the plate count is in the order of 20%, this difference is unlikely to be significant. Repeat experiments with sucrose and trehalose showed similar results, although the absolute survival percentages varied between experiments. Therefore, 10% sucrose was chosen for further development.

An overview of the various runs, carried out all in the same type of bioreactor, is shown in Table 4, indicating the manufacturing method, such as direct dilution of the culture in lyophilization buffer, concentration and diafiltration of the culture, followed by dilution with lyophilization buffer and concentration of the culture followed by dilution with lyophilization buffer.

The main reason to diafilter the BPZE1 drug substance was to reduce the salt content coming from the medium: 1.66 g/L NaCl and 0.765 g/L Tris, since the presence of these salts resulted in a slower lyophilization cycle than without the salts. However, in all drug substances that were concentrated and diafiltered some degree of clumping was observed (Table 4).

Thijs medium is chemically defined and consists of components that are all generally regarded as safe. Therefore, there is no need to remove these components from the BPZE1 vaccine from a quality perspective. Cultures that were either directly diluted with lyophilization buffer (Table 4, Runs 1a and 6b) or were concentrated and subsequently diluted with lyophilization buffer (Table 4, Run 7) did not show any sign of clumping directly after the harvest and just before filling. To comfortably meet the CFU target of the drug product, the final procedure chosen for further clinical development was to concentrate the culture to an OD_600_ of 5.0, followed by diluting the bacterial suspension 1:1 with cold lyophilization buffer (Table 4, Run 7).

The hold time between harvest and the start of lyophilization had a major influence on bacterial survival both before and after lyophilization. Runs 1 and 2 showed high post-lyophilization survival of 64% using 1:1 direct dilution of the culture with lyophilization buffer (Table 4, Run 1a), whereas the diafiltered cultures showed 46% and 47% survival (Table 4, Run 1b and Run 2). These drug products were lyophilized within 16 h after harvest and formulation, whereas all subsequent runs were lyophilized between 26 and 32 h after harvest (Table 4, footnotes 2 and 3, respectively). Samples of Runs 6b and 7 were tested for viability of the drug substance directly after formulation and after 48 h of storage at 4 °C. Both drug substances had lost approximately half the CFU. Therefore, the suspensions that were lyophilized 26 to 32 h after harvest will likely also have lost a considerable amount of CFU. This became apparent by comparing Run 1 with Run 6, which only differed in the duration of the hold time: Run 1 showed survival percentages of 46% to 64%, whereas Run 6 showed survival percentages of 8% and 18% (Table 4). Thus, the hold time duration prior to lyophilization had a significant impact on post-lyophilization survival.

### 3.3. Genetic Comparison of Liquid and Lyophilized BPZE1 Drug Products

The lyophilized drug product was compared to the liquid formulation stored at −70 °C to verify that the mutations introduced into the *B. pertussis* genome to generate BPZE1 were conserved, in particular the deletion of the *dnt* gene, the replacement of the *B. pertussis ampG* gene by the *Escherichia coli ampG* gene and the presence of the two mutated codons in the PT S1 subunit gene. The first two genetic modifications were verified by PCR, as described in [20]. The presence of the *E. coli ampG* gene was detected by the amplification of a 402-bp fragment corresponding to an internal fragment of the *E. coli ampG* gene. The two lyophilized BPZE1 drug products and the two liquid BPZE1 drug product controls yielded the expected 402-bp fragment, whereas this was not seen in the BPSM control sample (Figure 1A).

Conversely, a 659-bp fragment corresponding to the *B. pertussis ampG* gene was amplified in the BPSM control sample, but not in any of the BPZE1 drug products (Figure 1B), indicating that both the liquid and the lyophilized BPZE1 drug products lacked *B. pertussis ampG*, but contained *E. coli ampG*. The deletion of the *dnt* gene was shown by the amplification of a 1511-bp fragment resulting from a PCR using primers that flank the deleted *dnt* gene. The two lyophilized BPZE1 drug products and the two liquid BPZE1 drug product controls yielded the expected 1511-bp fragment, whereas this was not seen in the BPSM control sample (Figure 1C).

To verify the presence of the two mutated codons in the PT S1 gene, a quantitative PCR method was developed, which was able to detect one copy of the wild-type gene among 10^6^ copies of the mutated gene. For this purpose, 10^7^ copies of BPZE1 DNA and 10^7^ copies of BPZE1 DNA spiked with 10 copies of BPSM DNA were subjected to qPCR using BPSM- or BPZE1-specific oligonucleotides. 10^7^ copies of BPSM DNA served as a control. The threshold of positivity was set at 35 qPCR cycles. The lyophilized BPZE1 drug product and the liquid BPZE1 formulation showed indistinguishable amplification patterns, i.e., no amplification was observed with the BPSM-specific primers, whereas amplicons were detected with Cp values between 12.21 and 13.32 when using BPZE1-specific primers. In contrast, BPSM DNA was amplified with the BPSM-specific primers, but not with the BPZE1-specific primers, whereas spiked BPZE1 DNA was amplified with both primer pairs (Figure 2). These results indicate that BPZE1 had retained the codon modifications and that no reversion occurred at a frequency higher than 1/10^6^.

### 3.4. Microbiological Stability

The stability of the liquid BPZE1 drug products stored at −70 °C was followed up for two-year storage at −70 °C at three different formulations, 10^7^ (low dose), 10^8^ (middle dose) and 10^9^ CFU/dose (high dose). As shown in Figure 3A, the liquid BPZE1 formulation stored at −70 °C was stable for a minimum of two years at each dose tested.

Since the frozen liquid formulation proved to be stable for at least two years at −70 °C, a lyophilized drug product was expected to be stable for at least two years as well, either at −20 or 5 °C. There fore, the microbiological stability of the BPZE1 drug product formulated at 10^9^ CFU/dose (Table 4, Run 6b)was tested at −20 °C ± 10 °C, 5 °C ± 3 °C and 22.5 °C ± 2.5 °C. At all tested temperatures, the 10^9^ CFU/dose BPZE1 drug product met the CFU specification, even when stored at 22.5 °C ± 2.5 °C for at least two years (Figure 3B).

Whereas no sign of CFU loss was seen in the drug product stored at −20 °C ± 10 °C or 5 °C ± 3 °C, the drug product stored at 22.5 °C ± 2.5 °C showed some loss in viability during the first months of storage, but remained stable thereafter up to at least two years. Nevertheless, even in this case the CFU counts remained within specifications. The stability data of Run 7, which was produced by concentrating the culture and diluting it with lyophilization buffer, was similar to that of Run 6b, albeit at a higher CFU count due to the concentration step prior to adding the lyophilization buffer (data not shown).

### 3.5. Biological Stability

The biological stability of the lyophilized BPZE1 drug product (Table 4, Run 6b) was evaluated in two different mouse assays: an *in-vivo* colonization assay and a potency assay. In each of these assays the performance of the BPZE1 drug product stored at the various temperatures was compared with that of the original liquid formulation of BPZE1, stored at −70 °C.

To quantify the kinetics of *in-vivo* colonization, mice were intranasally inoculated with approximately 10^5^ CFU of reconstituted lyophilized BPZE1 drug product stored at different temperatures or the liquid BPZE1 drug product control. Three hours, one day and three days after administration, mice were sacrificed and CFU counts in the nasal homogenates were conducted. First, the effect of lyophilization and the composition of the lyophilization buffer was tested by comparing the liquid formulation with the lyophilized drug product immediately after lyophilization. As shown in Figure 4A, both formulations colonized the murine nasal cavity equally well, as there was no statistically significant difference between the liquid formulation and the lyophilized drug product.

The lyophilized drug product was then stored for two years at −20 °C ± 10 °C, 5 °C ± 3 °C or 22.5 °C ± 2.5 °C, and colonization kinetics were evaluated after six months (Figure 4B) and 24 months (Figure 4C) of storage and compared to those of the liquid formulation. Although after six months of storage, the material stored at −20 °C ± 10 °C adhered slightly better on day 0 and colonized faster one day after administration than the material stored at the other temperatures, this difference was no longer detected three days after administration (Figure 4B). However, after 24 months of storage, the lyophilized drug product stored at 5 °C ± 3 °C and 22.5 °C ± 2.5 °C adhered slightly less on day 0 and colonized slightly slower at both one and three days after administration than the drug product stored at −20 °C ± 10 °C (Figure 4C).

To evaluate the potency of the BPZE1 drug product after storage at different temperatures, mice were intranasally immunized with 10^5^ CFU of the reconstituted, lyophilized BPZE1 drug product or with the BPZE1 liquid drug product control, followed by an intranasal challenge with virulent BPBCTA1. Mice were sacrificed 3 h or seven days after the BPBCTA1 challenge to evaluate the bacterial load in the lungs. First, the liquid formulation was compared to the lyophilized drug product tested immediately after lyophilization. Both formulations protected mice equally well, as the CFU counts in the lungs decreased by two orders of magnitude between day 0 (3 h) and day 7 after challenge, whereas in the lungs of the non-vaccinated mice the bacterial load increased between day 0 and day 7 (Figure 5A).

Storage of the lyophilized drug product for six months at all temperatures tested did not affect the vaccine potency, as seven days after challenge unvaccinated mice carried approximately ten-fold more BPBCTA1 bacteria in their lungs than 3 h post-infection, whereas all vaccinated mice showed an approximately 100-fold reduction of CFU in their lungs, compared to those of the non-vaccinated controls (Figure 5B). No statistical difference was seen between mice immunized with the liquid BPZE1 drug product and those immunized with the lyophilized BPZE1 drug product, and no influence of the storage temperature could be detected. Thus, despite the slightly lower adherence on day 0 and the slower colonization of the mouse nasal cavity on day 1 by the lyophilized BPZE1 drug product stored at 5 °C ± 3 °C or 22.5 °C ± 2.5 °C compared to the product stored at −20 °C ± 10 °C, this had no effect on the lyophilized drug product’s ability to provide protection against a BPBCTA1 challenge, when stored for six months.

After 24 months of storage, the lyophilized drug products stored at 5 °C ± 3 °C or at 22.5 °C ± 2.5 °C showed a slight but significant decrease in potency, compared to the lyophilized drug product stored at −20 °C ± 10 °C (Figure 5C). However, compared to the non-vaccinated mice, those that had received the drug product stored at 5 °C ± 3 °C or at 22.5 °C ± 2.5 °C still showed an almost 1000-fold decrease in bacterial burden in the lungs.

Together, these data show that after storage of the lyophilized BPZE1 drug product between −20 °C ± 10 °C and 22.5 °C ± 2.5 °C for at least two years, the lyophilized BPZE1 maintained its ability to colonize the nasal cavity and its ability to protect mice from virulent *B. pertussis* challenge within specifications.

## 4. Discussion

In previous studies a single nasal administration of BPZE1 was shown to provide protection against a *B. pertussis* challenge in mice [8,10] and non-human primates [9], and was found to be safe, even in severely immunocompromised animals, such as IFN-γ receptor KO mice [22] and MyD88-deficient mice [19]. BPZE1 was also shown to be safe and immunogenic in two human phase I clinical trials [11,12]. Based on the safety and immunogenicity results of the phase Ib study, the 10^9^ CFU dose is being used in further clinical development of BPZE1. At this dose, 100% of study participants (12 out of 12) in the phase 1b study seroconverted [12]. Therefore, the development work presented here focused on the 10^9^ CFU dose.

All pre-clinical and clinical studies so far have been performed with a liquid formulation of BPZE1 that had to be stored at ≤−70 °C, a temperature providing stability for at least two years at 10^7^ CFU/mL, 10^8^ CFU/mL and 10^9^ CFU/mL (Figure 3). However, storage at −70 °C is incompatible with further clinical and commercial development. Here, we report that a lyophilized BPZE1 drug product was manufactured, that is stable for at least two years at −20 °C ± 10 °C, 5 °C ± 3 °C or 22.5 °C ± 2.5 °C.

Several product target attributes were formulated prior to initiating BPZE1 process development, as listed in Table 2. A post-lyophilization survival of 20% was targeted, as this was the survival percentage of the liquid BPZE1 drug product used in the phase I trials [11,12]. The target for the lyophilized BPZE1 drug product was that the CFU counts should remain between 0.2 and 4 × 10^9^ CFU/mL over at least a two-year storage period at 4 °C.

The survival after lyophilization of a live organism depends on the lyophilization cycle, lyophilization buffer and the physiological state of the organism prior to lyophilization. These parameters are likely interdependent. However, it became apparent that the survival of BPZE1 after lyophilization also depends on culture and harvest conditions. In particular, shear stress had a significant impact on post-lyophilization survival.

The critical importance of the hold time of the liquid bacterial suspension between harvest and start of lyophilization also became apparent during the actual production runs. Although initial runs, in which the start of lyophilization followed the harvest within 16 h, yielded 46% to 64% bacterial survival, hold times between 26 h and 32 h resulted in a reduction in survival to approximately 20%. Evaluating the survival after a hold time of 24 h to 48 h is particularly important for large-scale production, since harvesting, concentrating and formulating the bacterial suspension, and especially filling >200,000 vials per batch takes between 24 and 48 h.

The RMC of the drug product was consistently below 2.5%, which is generally compatible with long term stability at 5 °C or lower. However, the relation between temperature and post-lyophilization survival is determined by the Tg, which is the temperature at which the remaining water in the lyophilized product becomes mobile again, leading to accelerated loss of viability. A target Tg was set at ≥35 °C for logistical and supply chain reasons, since relatively brief exposure (from hours to several days) of the drug product to ambient, albeit controlled temperatures, does not significantly affect the drug product, as confirmed by the stability of the lyophilized drug product for two years at 22.5 °C ± 2.5 °C.

The manufacturing process for the lyophilized BPZE1 product did not affect the key molecular characteristics of the attenuated BPZE1 vaccine, i.e., the replacement of the *B. pertussis ampG* gene by that of *E. coli*, the deletion of the *dnt* gene, as assessed by specific PCRs, and the modifications of the PT S1 subunit gene that result in genetically inactivated PT, as assessed by a qPCR procedure, able to detect one putative reversion among 10^6^ genome equivalents.

Although the RMC and Tg are generally indicative of the expected stability, there is no substitute for real-time stability. Therefore, the lyophilized BPZE1 drug product was subjected to a real-time stability study at −20 °C ± 10 °C, 5 °C ± 3 °C and 22.5 °C ± 2.5 °C. The lyophilized BPZE1 drug product manufactured by direct dilution and by concentration and dilution showed that the drug product was stable when stored at −20 °C ± 10 °C, 5 °C ± 3 °C and 22.5 °C ± 2.5 °C over a period of at least 24 months, as CFU counts did not drop below the specification of 0.2–4 × 10^9^ CFU/mL during storage.

The biological activity of the organism, especially during the early steps in BPZE1 vaccination, such as attachment to and colonization of ciliated cells in the human nasopharynx, was expected to be similar between the liquid and lyophilized drug products. Adherence and colonization kinetics were evaluated in mice using a liquid formulation stored at −70 °C, containing 5% sucrose in PBS [11,12]. The phase Ib clinical study showed that this liquid formulation led to detectable colonization of >80% of the subjects [12], even though PBS is hypertonic as compared to the salinity of the respiratory tract. The decrease in salinity from PBS + 5% sucrose in the liquid formulation to the lower osmolarity of the Thijs medium + 10% sucrose did not affect the adherence or the colonization of the murine nasal cavity. Although two-year storage at 5 °C ± 3 °C or 22.5 °C ± 2.5 °C appeared to slightly but significantly reduce the adherence and the speed of colonization, this had only a minimal effect on vaccine potency, since the lyophilized BPZE1 drug product stored for 24 months at any of the temperatures tested still provided protection, i.e., a more than 100-fold reduction in bacterial load compared to non-vaccinated controls seven days after challenge.

Although the described manufacturing process is sufficiently robust to be used for further clinical and commercial development, there are possibilities to further increase drug product viability after lyophilization by decreasing the hold time and increasing the production capacity at a large scale. These include lyophilization in bulk, followed by milling and formulating the lyophilized cake with excipients to the desired CFU count and dispensing the formulated powder in a suitable drug product container, which are currently being explored.

## 5. Conclusions

In conclusion, we describe here a procedure that yields a lyophilized BPZE1 drug product, which is stable for up to at least 24 months of storage at −20 °C ± 10 °C, 5 °C ± 3 °C and 22.5 °C ± 2.5 °C. Microbiological stability, as well as *in-vivo* adherence and vaccine potency, were within specifications after 24 months of storage at either temperature. This lyophilized drug product is therefore suitable for further clinical development and commercialization.

## Figures and Tables

**Figure 1 vaccines-08-00523-f001:**
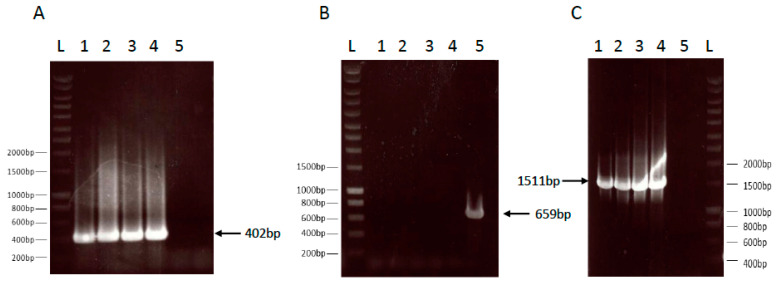
PCR analyses of the *ampG* and *dnt* loci of the lyophilized BPZE1 drug product compared to the liquid drug product. *E. coli ampG* (panel **A**), *B. pertussis ampG* (panel **B**) and the *B. pertussis dnt* flanking regions (panel **C**) of two lots of the liquid BPZE1 drug product (lanes 1 and 2) and two lots of the lyophilized BPZE1 drug product (lanes 3 and 4), as well as a BPSM control (lanes 5) were amplified by PCR using the appropriate primers listed in Table 1. Lanes L contain the molecular size markers with relevant sizes in bp indicated in the margins. The arrows point to the expected amplicons of 402 bp (panel **A**), 659 bp (panel **B**) and 1511 bp (panel **C**) for *E. coli ampG*, *B. pertussis ampG* and the *B. pertussis dnt* flanking regions, respectively.

**Figure 2 vaccines-08-00523-f002:**
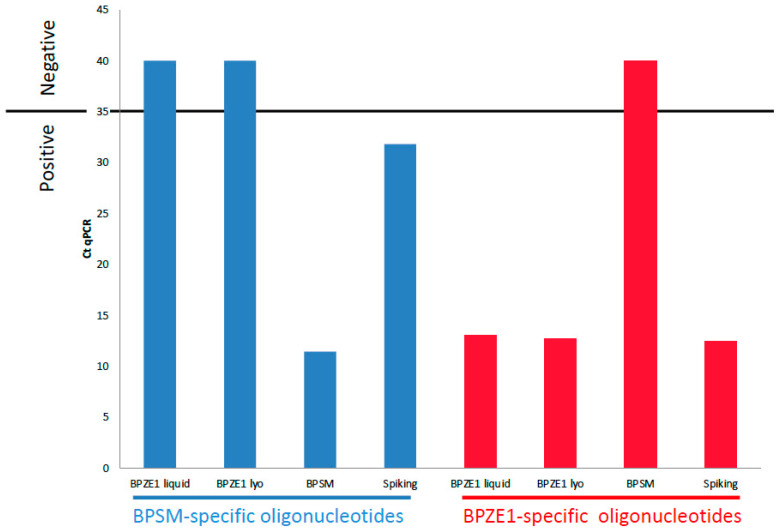
q-PCR amplification of the PT S1 subunit-coding DNA. The S1 subunit gene of the liquid BPZE1 drug product (BPZE1 liquid), the lyophilized BPZE1 drug product (BPZE1 lyo), BPSM and BPSM-spiked lyophilized BPZE1 (Spiking) was amplified by PCR using the BPSM-specific (blue) or BPZE1-specific primers (red), as listed in Table 1. The horizontal line indicates the cut-off in numbers of amplification cycles (Ct qPCR) indicating whether a sample is considered positive (≤35 cycles) or negative (≥35 cycles).

**Figure 3 vaccines-08-00523-f003:**
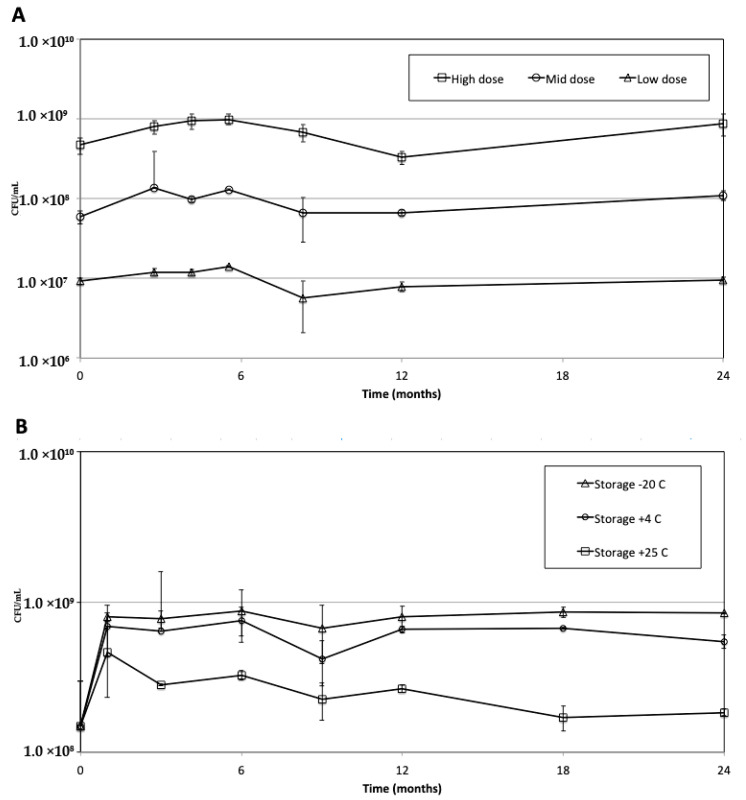
Microbiological stability of the BPZE1 drug products over time. (**A**) The liquid BPZE1 drug product at 10^7^ CFU/dose (triangles, low dose), 10^8^ CFU/dose (circles, middle dose) and 10^9^ CFU/dose (squares, high dose) was stored at −70 °C for two years, and CFU counts were conducted at the indicated time points. (**B**). Microbiological stability of the lyophilized BPZE1 drug product over time. The lyophilized BPZE1 drug product at 10^9^ CFU/dose was stored at −20 °C ± 10 °C (triangles), 5 °C ± 3 °C (circles) and 22.5 °C ± 2.5 °C (squares) for two years, and CFU counts were conducted at the indicated time points. Vertical lines represent standard deviations.

**Figure 4 vaccines-08-00523-f004:**
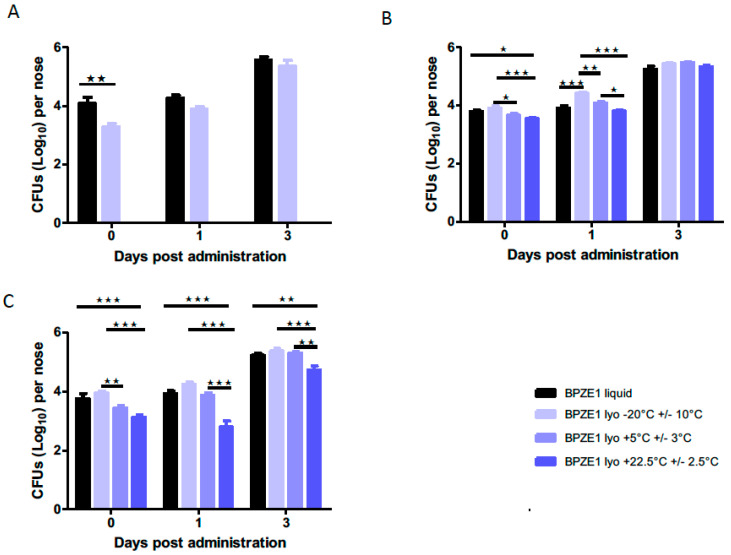
*In-vivo* colonization kinetics of the lyophilized BPZE1 drug product compared to the liquid drug product. BALB/c mice (n = 5 per group and per time point) were inoculated intranasally with 10^5^ CFU of the liquid BPZE1 drug product (black bars) or the reconstituted lyophilized BPZE1 drug product (blue bars) and sacrificed 3 h (day 0), 1 or 3 days thereafter to evaluate the CFU numbers in the nasal homogenates. (**A**) Comparison of the CFU counts of the liquid BPZE1 drug product with those of the lyophilized BPZE1 drug product reconstituted and administered immediately after lyophilization. (**B**) Comparison of the CFU counts of the liquid BPZE1 drug product with those of the lyophilized BPZE1 drug product reconstituted 6 months after storage at −20 °C ± 10 °C (light blue lines), 5 °C ± 3 °C (middle blue lines) or 22.5 °C ± 2.5 °C (dark blue lines). (**C**) Comparison of the CFU counts of the liquid BPZE1 drug product with those of the lyophilized BPZE1 drug product reconstituted 24 months after storage at −20 °C ± 10 °C (light blue lines), 5 °C ± 3 °C (middle blue lines) or 22.5 °C ± 2.5 °C (dark blue lines). The results are expressed as means +/– SEM. *, *p* < 0.05; **, *p* < 0.01; ***, *p* < 0.005; ns, not significant.

**Figure 5 vaccines-08-00523-f005:**
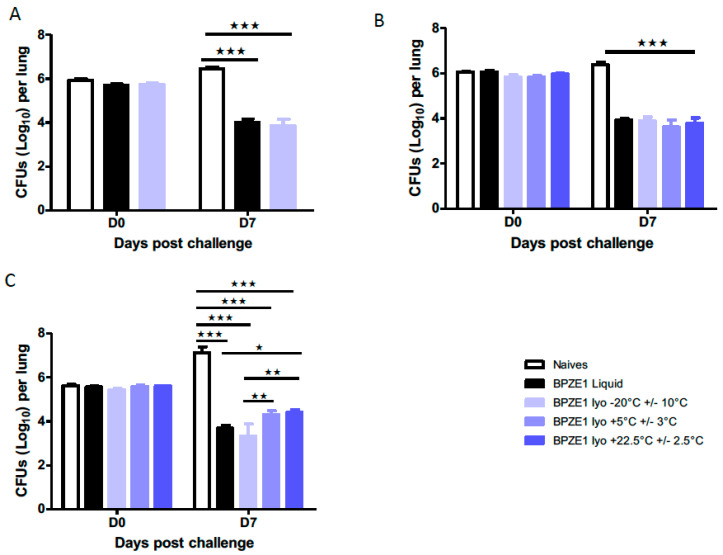
Potency of the lyophilized BPZE1 drug product compared to the liquid drug product. BALB/c mice (n = 5 per group and per time point) were intranasally vaccinated with 10^5^ CFU of the liquid BPZE1 drug product (black bars) or the reconstituted lyophilized BPZE1 drug product (blue bars), or received PBS as a mock control (white bars). Four weeks later, the mice were challenged intranasally with 10^6^ CFU of virulent BPBCTA1. CFU counts in the lungs were conducted 3 h (D0) and 7 days (D7) post-challenge. (**A**) Comparison of the potency of the liquid BPZE1 drug product with that of the lyophilized BPZE1 drug product reconstituted and administered immediately after lyophilization. (**B**) Comparison of potency of the liquid BPZE1 drug product with that of the lyophilized BPZE1 drug product reconstituted 6 months after storage at −20 °C ± 10 °C (light blue lines), 5 °C ± 3 °C (middle blue lines) or 22.5 °C ± 2.5 °C (dark blue lines). (**C**) Comparison of the potency of the liquid BPZE1 drug product with that of the lyophilized BPZE1 drug product reconstituted 24 months after storage at −20 °C ± 10 °C (light blue lines), 5 °C ± 3 °C (middle blue lines) or 22.5 °C ± 2.5 °C (dark blue lines). The results are expressed as means +/– SEM. *, *p* < 0.05; **, *p* < 0.01; ***, *p* < 0.005.

**Table 1 vaccines-08-00523-t001:** Sequences of primers used for PCR.

*Escherichia coli ampG*	Forward: 5′-ATG TGC TTC CGG CAG AAG AA-3′ Reverse: 5’-CAA GCG TTT TGT TAA CCA CG-3’
*B. pertussis ampG*	Forward: 5′- TCG CAG GAC ATC GCC TTC GA-3′ Reverse: 5’-ATC AGC AGC GCC ACG AAG GA-3’
*B. pertussis dnt*	Forward: 5′-TAT AGA ATT CGC TCG GTT CGC TGG TCA AGG-3′ Reverse: 5’-TAT AAA GCT TCT CAT GCA CGC CGG CTT CTC-3’
BPZE1-specific primers	Forward: 5′-CTC CCG CCA CCG TAT ACA AG-3′ Reverse: 5’-GCG CCG GTG TGC CAG ATA GC-3’
BPSM-specific primers	Forward: 5′-CTC CCG CCA CCG TAT ACC GC-3′ Reverse: 5’-GCG CCG GTG TGC CAG ATA TT-3’

**Table 2 vaccines-08-00523-t002:** Target drug product attributes for the lyophilized BPZE1 drug product.

Attribute	Target Specification	Method
Hold time prior to lyophilization	24–48 h	Plate count on Bordet–Gengou plates
Appearance upon reconstitution	opaque liquid, no visible clumping	visual inspection USP<790>
Shelf life post lyophilization	0.2–4 × 10^9^ CFU/mL for ≥2 year	Plate count on Bordet–Gengou plates
Survival post lyophilization	≥20%	Plate count on Bordet–Gengou plates
Residual moisture content (RMC)	≤2.5%	Karl Fischer, USP<921> [16]
Glass transition temperature (Tg)	≥35 °C	Differential scanning calorimetry
Adherence and colonization of murine nasal cavity	comparable to liquid phase Ib formulation	Homogenization of murine nasal cavity followed by plating the homogenate on Bordet–Gengou plates
Potency assay	Reduction in bacterial burden of ≥100-fold compared to controls	Homogenization of murine lungs 3 h and 7 days after BPBCTA1 challenge followed by plating the homogenate on Bordet–Gengou plates

**Table 3 vaccines-08-00523-t003:** Residual moisture content, glass transition temperature and bacterial survival as a function of lyophilization buffer conditions.

Trehalose	Sucrose	HES ^1^	Na-Glutamate	RMC (%)	Tg (°C)	Survival (%) ^2^
-	5%	-	-	1.0	43	44
-	10%	-	-	1.8	36	38
-	5%	5%	-	0.2	48	25
-	10%	7%	-	0.3	46	44
5%	-	-	-	0.6	65	24
10%	-	-	-	0.4	53	33
5%	-	5%	-	0.2	54	24
10%	-	7%	-	0.3	54	26
5%	-	7%	1%	0.3	54	22
10%	-	7%	2%	0.3	57	31

^1^ HES, hydroxy ethyl starch; RMC, residual moisture content; Tg, glass transition temperature. ^2^ Survival is expressed as percentage of colony forming units (CFU) comparing the pre- and post-lyophilization content of the vials.

**Table 4 vaccines-08-00523-t004:** Overview of the various runs carried out in a 50-L single-use bioreactor with 20 L medium, comparing different harvest methods.

Batch:		Run 1 ^1^		Run 2 ^2^	Run 3 ^3^	Run 4 ^3^	Run 5 ^3^	Run 6 ^3^		Run 7 ^3^
		1a	1b					6a	6b	
Manufactured by:		direct dilution ^4^		concentration and diafiltration ^5^					direct dilution ^4^	concentration ^6^
**Test**	**Proposed Specification**	**Results**								
Hold time	24–48 h	6	6	16	28	31	26	28	28	32
Homogeneity	Homogeneous suspension	not tested	not tested	minor clumping	minor clumping	minor clumping	severe clumping	severe clumping	pass	pass
Plate count pre-lyophilization (×10^9^)	0.4–8.0 CFU/mL	1.1	2.4	6.8	3.2	1.8	7.7	2.4	2.2	8.3
Plate count post lyophilization (×10^9^)	0.2–4.0 CFU/mL	0.7	1.1	3.2	0.6	0.3	0.4	0.2	0.4	1.9
Survival %	-	64	46	47	19	17	5	8	18	23

^1^ Run 1 was filled <50 vials, lyophilization was started with <6 h hold time. ^2^ Run 2 was filled <700 vials, lyophilization was started with <16 h hold time. ^3^ Runs 3 to 7 were harvested, lyophilized in 2000 to 7000 vials per formulation and lyophilization was initiated after between 24 and 36 h hold time. ^4^ Direct dilution: dilution of the culture 1:1 with lyophilization buffer. ^5^ Concentration and diafiltration: concentration of the culture followed by diafiltration and 1:1 dilution with lyophilization buffer. ^6^ Concentration: concentration of the culture followed by 1:1 dilution with lyophilization buffer.

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
