# Peer review of "Manufacture of a Stable Lyophilized Formulation of the Live Attenuated Pertussis Vaccine BPZE1"

_vaccines, 2020, doi:10.3390/vaccines8030523_

Round 1

Reviewer 1 Report

The authors describes in the manuscript their efforts to manufacture a stable lyophilized formulation of the live attenuated pertussis vaccine BPZE1. This is a huge challenge that would impact the success in the commercialization of this vaccine. Many hurdles are presented in this apparently simple development. However, cell clamping, viability and efficacy are the main barriers to be overcome.

I would like to address some comments on the manuscript:

1) The first step in the industrial development of this vaccine was the develop the BPZE1 drug substance. B. pertussis produces a number of virulence factors in a temporal conditions that can lead to biofilm formation and cell clumping. Therefore, establishing the best condition to grow B. pertussis without clumping and affecting its immunogenicity is an important compromise. In this sense, the authors have used the Thijs medium. However, usually all the manufactures adopt the use of Stainer-Scholte synthetic broth medium. There is no explanation why the authors chose the Thijs medium instead of Stainer-Scholte medium or if it is based on Stainer-Scholte medium. This reviewer knows that this information is described by Thalen et al, 2006, however, for most of the readers, this relevant information may not be of their knowledge; 

2) The results summarized in Table 4 is very complex and very difficult to follow. There are no clear explanation on the differences between the runs and why they were established in a clear manner. The runs were just partially described in the results section without a proper rational experimental description for the reader to follow;

3) Concerning the stability of the Lyophilized BPZE1, it would be important to evaluate its stability when manufactured from a frozen liquid BPZE1 kept at -70°C for a short term and after 18 months of storage. This is an important information to evaluate the limit of use of frozen liquid BPZE1 for the manufacture of Lyo BPZE1;

4) Please, describe in details the type of vial used for lyophylization, the stopper used and the final volume filled in each vial (at 0.1-4.0 x 10E9 CFU/ml). When used for animal experimentation, in what the cells were resuspended as well as the volume used? After resuspension, how the vaccine was administered to the animals? Directly  from the resuspension? After dilution to reach 10E5/20 ul? The authors should give more details in all these steps that will be important for the understanding of how these procedures will be translated for human use.

5) In these conditions, how many vaccine doses were produced, considering 0.1-4,0 x 10E9 CFU/ml?

It is my opinion that this is an important step to bring this important vaccine for commercialization, although some points as pointed above should be clarified.

Author Response

The authors describes in the manuscript their efforts to manufacture a stable lyophilized formulation of the live attenuated pertussis vaccine BPZE1. This is a huge challenge that would impact the success in the commercialization of this vaccine. Many hurdles are presented in this apparently simple development. However, cell clamping, viability and efficacy are the main barriers to be overcome.

I would like to address some comments on the manuscript:

1) The first step in the industrial development of this vaccine was the develop the BPZE1 drug substance. B. pertussis produces a number of virulence factors in a temporal conditions that can lead to biofilm formation and cell clumping. Therefore, establishing the best condition to grow B. pertussis without clumping and affecting its immunogenicity is an important compromise. In this sense, the authors have used the Thijs medium. However, usually all the manufactures adopt the use of Stainer-Scholte synthetic broth medium. There is no explanation why the authors chose the Thijs medium instead of Stainer-Scholte medium or if it is based on Stainer-Scholte medium. This reviewer knows that this information is described by Thalen et al, 2006, however, for most of the readers, this relevant information may not be of their knowledge; 

Response: The reason to use Thijs medium instead of Stainer Scholte medium was because Thijs medium is optimized for virulence gene expression, which is now mentioned in lines 69-70, and we added the reference to the Stainer Scholte Medium (Ref. #17) in the revised version.

2) The results summarized in Table 4 is very complex and very difficult to follow. There are no clear explanation on the differences between the runs and why they were established in a clear manner. The runs were just partially described in the results section without a proper rational experimental description for the reader to follow;

Response: Table 4 has been modified in the revised version, and we hope that it is now more legible, and the three paragraphs describing table 4 have been modified, emphasizing the rationale behind the runs.

3) Concerning the stability of the Lyophilized BPZE1, it would be important to evaluate its stability when manufactured from a frozen liquid BPZE1 kept at -70°C for a short term and after 18 months of storage. This is an important information to evaluate the limit of use of frozen liquid BPZE1 for the manufacture of Lyo BPZE1;

Response: The clinical material for the phase Ib study in 2015 was made with a Working Cell Bank that was less than 2 months old when the clinical material was made. The material shown in this study, was made in 2018, i.e. 3 years after the WCB was made. Both products are directly compared in the mouse potency and adherence assay and behave very similar, as described in Figures 5 and 6 (now Figures 4 and 5).

4) Please, describe in details the type of vial used for lyophylization, the stopper used and the final volume filled in each vial (at 0.1-4.0 x 10E9 CFU/ml). When used for animal experimentation, in what the cells were resuspended as well as the volume used? After resuspension, how the vaccine was administered to the animals? Directly  from the resuspension? After dilution to reach 10E5/20 ul? The authors should give more details in all these steps that will be important for the understanding of how these procedures will be translated for human use.

Response: in the materials & methods section 2.3 the vial and stopper are described (DIN 2R vial, 13 mm bromobutyl lyophilization stopper), the filling volume of 1 mL was added and now reads: The resulting formulated drug substance was then filled in 1-mL aliquotsin a DIN 2R vials with a 13 mm bromobutyl lyophilization stopper.... (see lines 85 - 87).

For the animal experiments, the lyophilized cells were resuspended in distilled water and diluted with PBS to reach the desired concentration of bacteria per 20 µl. This is now specified in lines 110 - 112 of the revised version.

5) In these conditions, how many vaccine doses were produced, considering 0.1-4,0 x 10E9 CFU/ml?

Response: in the footnote #3 to Table 4 the number of vials per harvest is specified, i.e. 2000 to 7000, depending on whether one or more sub-batches were produced, i.e Runs 1 and 6 were formulated to 2 sub-batches each.

It is my opinion that this is an important step to bring this important vaccine for commercialization, although some points as pointed above should be clarified.

Reviewer 2 Report

This article by Thalen et al. describe the manufacturing process and quality control validation of a stable lyophilized formulation of a live-attenuated vaccine against B. pertussis.

To my opinion, this article deserve publication after revision.

General comments: 

English reviewing could be necessary, including typo correction (see line 9 : duplication of correspondence for example). Reference have to be indicated at the end of the sentence, except for justified sentence.

Prefer pasive form in the manuscript.

Paragraph 2.6 and 2.7 :

For manufacturer's identifications, please add city and country, for line 121 : BufferB1, please add complementary identification.

Please give details of the number of mice tested per condition.

Table 3:

Please modify the table to enlarge different columns.

Have the condition of Threhalose 10%/HES5%; and Threhalose5% or 10%/HES7% and Na-glutamate1% or 2% been tested? if not justify.

Table 4

Could authors justify the duplication of th Run 1 and 6 in the legend of the table?

Line 200 : please give details about the data not shown has this experiment could be very interesting.

Figure 2 : These data could be easily understandable in a table than in a figure. Please modify.

Figure 3 and 4 could be reunited in a unique figure (or in a figure with A and B). Could the authors please modify the line types, so the manuscript could be printed in black and white?

Author Response

This article by Thalen et al. describe the manufacturing process and quality control validation of a stable lyophilized formulation of a live-attenuated vaccine against B. pertussis.

To my opinion, this article deserve publication after revision.

General comments: 

English reviewing could be necessary, including typo correction (see line 9 : duplication of correspondence for example). Reference have to be indicated at the end of the sentence, except for justified sentence.

Prefer pasive form in the manuscript.

Response: the manuscript has been re-reviewed independently by 2 native English speakers. The typos have been corrected, and where appropriate, the passive form is used. Most references are indicated at the end of the sentence, except when a reference refers to a particular part of the sentence only, and another reference refers to another part of the same sentence (e.g. lines 64 and 65, or lines 69 and 70).

Paragraph 2.6 and 2.7 :

For manufacturer's identifications, please add city and country, for line 121 : BufferB1, please add complementary identification.

Response: the manufacturer’s indications have now been added in lines 108 and 109, and in lines 124, 125 and 128 of the revised version.

Please give details of the number of mice tested per condition.

Response: the numbers of mice have now been added in the figure legends of figures 5 and 6 (now figures 4 and 5 of the revised version) (see lines 328 and 360).

Table 3:

Please modify the table to enlarge different columns.

Response: Table 3 has been modified so that component headings fit in the respective cells.

Have the condition of Threhalose 10%/HES5%; and Threhalose5% or 10%/HES7% and Na-glutamate1% or 2% been tested? if not justify.

Response: Table 3 shows that adding HES to either Trehalose or sucrose always yielded the same or lower survival percentages than without HES. The only exception is 10% Sucrose + 7% HES which yielded 44% survival, which is slightly higher than the 38% survival in the formulation with 10% Sucrose alone. The difference between the 2 formulations is small enough to be attributed to the measuring error of the plate count determination. The same consideration holds true for the addition of Na-glutamate into the formulation: the 3 component formulations yielded a similar or lower result than the single component formulation, indicating that neither HES nor Na-glutamate contributes to post lyophilization survival. To address the referee's point, the corresponding paragraph has been modified in lines 185-189 to:

The addition of HES, MSG or both to 5 or 10% of trehalose or sucrose appeared to reduce rather than enhance post-lyophilization survival, except for the addition of 7% HES to 10% sucrose which yielded 44% survival compared to 38% survival using sucrose by itself. However, since the variability of the plate count is in the order of 20%, this difference is unlikely to be significant.

Table 4

Could authors justify the duplication of th Run 1 and 6 in the legend of the table?

Response: the difference between Run 1 and Run 6 is the hold time between the culture harvest and lyophilization, for Run 1 the hold time was less than 6 hours (footnote #1 to Table 4) while for Run 6 the hold time was >24 hours (footnote #3 to Table 4). The impact of the length of the hold time is apparent since the survival in run 1 is between 46 and 64% while the survival in run 6 is between 8 and 18%. In order to make this point more clearly the last phrases of the corresponding paragraph have been modified in lines 226-234 to read:

Runs 6b and 7 were tested for viability of the drug substance directly after formulation and after 48 hours of storage at +4°C. Both drug substances had lost approximately half the CFU. Therefore, the suspensions that were lyophilized 26 to 32 hours after harvest will likely also have lost a considerable amount of CFU. This became apparent when comparing Run 1 with Run 6, which only differed in the duration of the hold time: Run 1 showed survival percentages of 46 to 64%, while Run 6 showed survival percentage of 8 and 18% (Table 4). Thus, the hold time duration prior to lyophilization had a significant impact on post-lyophilization survival.

Line 200 : please give details about the data not shown has this experiment could be very interesting.

Response: the data referred to was generated in 8 L cultures which showed significantly lower survival rates than those in the 20 L bioreactor runs shown in Table 4. Therefore, head to head comparison with the 20 L runs would require significant explanation and framing which we felt was not contributing to the narrative. We therefore decided to delete this sentence in the revised version (see lines 206-208).

Figure 2 : These data could be easily understandable in a table than in a figure. Please modify.

Response: we feel that a figure visualizes the point much better than a table. We therefore prefer maintaining this figure. However, if the editor strongly feels that we should change this into a table, we would of course be prepared to do so. 

Figure 3 and 4 could be reunited in a unique figure (or in a figure with A and B). Could the authors please modify the line types, so the manuscript could be printed in black and white?

Response: Figures 3 and 4 have now been combined in the revised version, and the line types have been modified as requested by the referee. 

Reviewer 3 Report

In the current manuscript, Thalen et al. present sound data on their efforts to produce a lyophilized, live-attenuated pertussis vaccine. The experiments are described in great detail, and the results are adequately presented. I have only some minor comments:

Lines 218-220 "both drug substances had lost approximately half the CFU, which explains the relatively poor survival of 18 and 23% ..." This sentence does not seem very clear to me. When was the pre-lyophilization count determined, was it directly after harvest or directly before lyophilization? If both counts were performed, as I assume from the statement in the text, they could both be shown in the table where applicable.

Figures 3 and 4 show the stability of the liquid and lyophilized BPZE1 as mean values of triplicates of 3 dilutions. Standard deviation should be included instead of showing only the mean.

Figure 5 shows the efficacy of in vivo colonization of the lyophilized product. While the data looks convincing, there is no information on the number of mice that were analyzed, and the number of experiments the data were obtained from? The same applies to Figure 6.

The discussion is mostly a reiteration of the results, this should be modified.

Author Response

In the current manuscript, Thalen et al. present sound data on their efforts to produce a lyophilized, live-attenuated pertussis vaccine. The experiments are described in great detail, and the results are adequately presented. I have only some minor comments:

Lines 218-220 "both drug substances had lost approximately half the CFU, which explains the relatively poor survival of 18 and 23% ..." This sentence does not seem very clear to me. When was the pre-lyophilization count determined, was it directly after harvest or directly before lyophilization? If both counts were performed, as I assume from the statement in the text, they could both be shown in the table where applicable.

Response: The referee's point has been addressed in the paragraph by modifying it in lines 220-234 to read:

 The hold time between harvest and the start of lyophilization had a major impact on bacterial survival both before and after lyophilization. Runs 1 and 2 showed high post lyophilization survival of 64% using 1:1 direct dilution of the culture with lyophilization buffer (Table 4, Run 1a), while the diafiltered cultures showed 46% and 47% survival (Table 4, Run 1b and Run 2). These drug products were lyophilized within 16 hours after harvest and formulation, while all subsequent runs were lyophilized between 26 and 32 hours after harvest (Table 4, footnotes 2 and 3, respectively). Runs 6b and 7 were tested for viability of the drug substance directly after formulation and after 48 hours of storage at +4°C. Both drug substances had lost approximately half the CFU. Therefore, the suspensions that were lyophilized 26 to 32 hours after harvest will likely also have lost a considerable amount of CFUs. This became apparent by comparing Run 1 with Run 6, which only differed in the duration of the hold time: Run 1 showed survival percentages of 46 to 64%, while Run 6 showed survival percentages of 8 and 18% (Table 4). Thus, the hold time duration prior to lyophilization had a significant impact on post-lyophilization survival.

Figures 3 and 4 show the stability of the liquid and lyophilized BPZE1 as mean values of triplicates of 3 dilutions. Standard deviation should be included instead of showing only the mean.

Response: Figures 3 and 4 have been combined into a single figure, and standard deviations have been included in the revised version.

Figure 5 shows the efficacy of in vivo colonization of the lyophilized product. While the data looks convincing, there is no information on the number of mice that were analyzed, and the number of experiments the data were obtained from? The same applies to Figure 6.

Response: the numbers of mice have now been added in the figure legends of figures 5 and 6 of the revised version (see lines 328 and 360).

The discussion is mostly a reiteration of the results, this should be modified.

Response: We have tried to limit the reiteration of the results to a minimum, but some key data have to be included into the discussion to make the discussion understandable and establish the link between the data and the points to be discussed. We hope this is acceptable.